# The Tropical Basin Interaction Model Intercomparison Project (TBIMIP)

Ingo Richter[1], Ping Chang[2], Ping-Gin Chiu[3], Gokhan Danabasoglu[4], Takeshi Doi[1], Dietmar Dommenget[5], Guillaume Gastineau[6], Aixue Hu[4], Takahito Kataoka[7], Noel S. Keenlyside[3,8], Fred Kucharski[9], Yuko M. Okumura[10], Wonsun Park[11], Malte F. Stuecker[12], Andréa S. Taschetto[13], Chunzai Wang[14], Stephen G. Yeager[4], Sang-Wook Yeh[15]

[1]Research Institute for Value-Added-Information Generation, Japan Agency for Marine-Earth Science and Technology, Yokohama, 236-0001, Japan
[2]Department of Oceanography, Texas A&M University, College Station, TX, USA
[3]Geophysical Institute, University of Bergen and Bjerknes Centre for Climate Research, Bergen, 5007, Norway
[4]Climate and Global Dynamics Laboratory, US National Science Foundation National Center for Atmospheric Research, Boulder, CO, USA
[5]ARC Centre of Excellence for Climate Extremes, School of Earth Atmosphere and Environment, Monash University, Clayton, VIC, 3800, Australia
[6]UMR LOCEAN, Sorbonne Université/CNRS/IRD/MNHN, Paris, France
[7]Research Center for Environmental Modeling and Application, Japan Agency for Marine-Earth Science and Technology, Yokohama, 236-0001, Japan
[8]Nansen Environmental and Remote Sensing Center, Bergen, 5007, Norway
[9]Earth System Physics, Abdus Salam International Centre for Theoretical Physics, Trieste, Italy
[10]Institute for Geophysics, Jackson School of Geosciences, University of Texas at Austin, Austin, TX, USA
[11]IBS Center for Climate Physics and Department of Climate System, Pusan National University, Korea
[12]Department of Oceanography and International Pacific Research Center, University of Hawai'i at Mānoa, Honolulu, HI, USA
[13]Climate Change Research Centre and ARC Centre of Excellence for the 21st Century Weather, University of New South Wales, Sydney, Australia
[14]State Key Laboratory of Tropical Oceanography, Global Ocean and Climate Research Center, Guangdong Key Laboratory of Ocean Remote Sensing, South China Sea Institute of Oceanology, Chinese Academy of Sciences, Guangzhou, China
[15]Department of Marine Sciences and Convergent Engineering, Hanyang University, Ansan, South Korea

*Correspondence to*: Ingo Richter (richter@jamstec.go.jp)

**Abstract.** Large-scale interaction among the three tropical ocean basins is an area of intense research that is often conducted through experimentation with numerical models. A common problem is that modelling groups use different experimental setups, which makes it difficult to compare results and to delineate the role of model biases from differences in experimental setups. To address this issue, an experimental protocol for examining interaction among the tropical basins is introduced. The tropical basin interaction model intercomparison project (TBIMIP) consists of experiments in which sea surface temperatures (SSTs) are prescribed to follow observed values in selected basins. There are two types of experiments. One type, called standard pacemaker, consists of simulations in which SSTs are restored to observations in selected basins during a historical simulation. The other type, called pacemaker hindcast, consists of seasonal hindcast simulations in which SSTs are restored to observations during the 12-month forecast periods. TBIMIP is coordinated by the Climate and Ocean - Variability,

Predictability, and Change (CLIVAR) Research Focus on Tropical Basin Interaction. The datasets from the model simulations
will be made available to the community to facilitate and stimulate research on tropical basin interaction and its role in
seasonal-to-decadal variability and climate change.

## 1 Introduction

Interaction among the tropical basins on interannual to decadal timescales has seen increased interest in recent decades. This
is partly due to the growing awareness that this interaction substantially influences variability in all three tropical basins (Cai
et al., 2019; Wang, 2019) and that it may also shape the way the climate system reacts to radiative forcing, particularly that
associated with changing greenhouse gas concentrations (Kosaka and Xie, 2013; Li et al., 2016). Furthermore, there is evidence
that the linkages among the three tropical basins will change under global warming, leading to the emergence of new processes
in the climate system, such as the tropical Atlantic influence on El Niño-Southern Oscillation (ENSO; Rodriguez-Fonseca et
al., 2009; Martin-Rey et al., 2014; Polo et al., 2015; Wang et al., 2024a), or that of the Indian Ocean on ENSO (Wang et al.,
2024b).

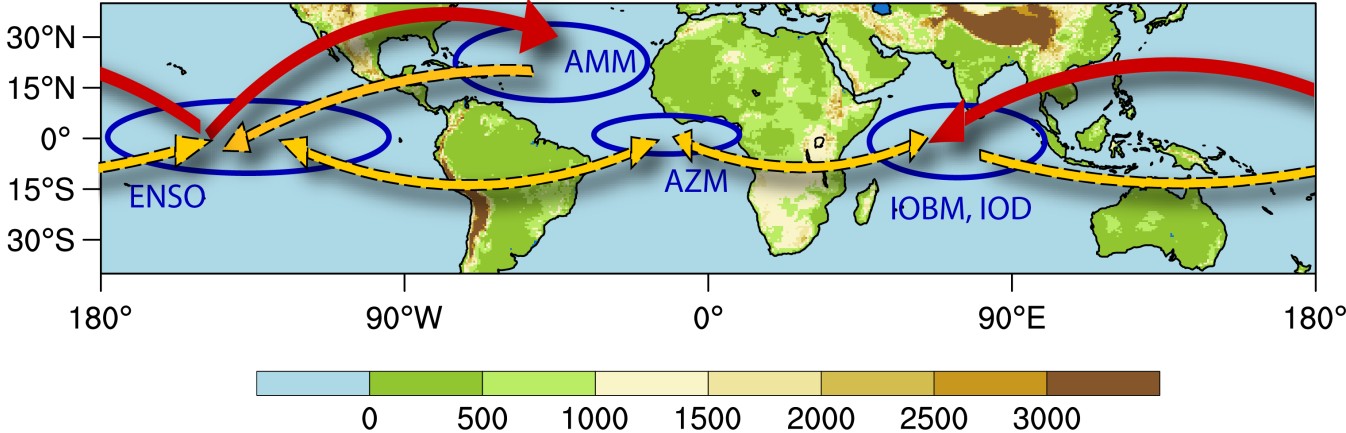

**Figure 1.    Schematic illustrating the interaction of selected tropical variability patterns, namely ENSO (El Niño-Southern**
**Oscillation), AMM (Atlantic Meridional Mode), AZM (Atlantic Zonal Mode), IOD (Indian Ocean Dipole), and IOBM (Indian Ocean**
**Basin Mode). The arrows illustrate the directionality of the influence and are not necessarily representative of the actual interaction**
**pathways. The AZM-to-ENSO influence, e.g., could be through atmospheric equatorial Rossby waves, as suggested by the arrow, or**
**through atmospheric equatorial Kelvin waves (not indicated). The solid red arrows show well-established influences, while the**
**dashed yellow arrows show influences that are under debate or inconsistent. The shading shows topographic heights (m) from the**
**Earth topography five-minute grid (ETOPO5), with ocean areas set to zero.**
Research on interbasin interaction has undergone several phases. In the 1970s and 1980s, many researchers focused on
understanding the mechanisms of ENSO in the tropical Pacific and the air-sea coupling that underlies it (e.g., Bjerknes, 1969;
McCreary, 1976; Rasmusson and Carpenter, 1982; McCreary and Anderson, 1984; Philander, 1985; Zebiak and Cane, 1987).
Over time, there was increasing interest in how ENSO influences other terrestrial and oceanic regions around the world (e.g.,
Bjerknes, 1969; Horel and Wallace, 1981; Karoly, 1989; Kiladis and Diaz, 1989; Enfield and Mestas-Nuñez, 1999; Klein et

al., 1999; Diaz et al., 2001; Alexander et al., 2002). During this stage, the focus was on remote influences from the tropical Pacific to other regions. At the same time, other tropical ocean regions received increasing attention, which led to the discovery and analysis of other tropical variability patterns, such as the Atlantic Zonal Mode (AZM; Moore et al., 1978; Hastenrath and Heller, 1977; Merle, 1980; reviews by Lübbecke et al., 2018; Richter and Tokinaga, 2021), the Indian Ocean Basin Mode (IOBM; Chambers et al., 1999; review by Schott et al., 2009), and the Indian Ocean Dipole (IOD; Saji et al., 1999; Webster et al., 1999; review by Schott et al., 2009). Several variability patterns in the subtropics also became more prominent, such as the Atlantic Meridional Mode (AMM; Hastenrath and Heller, 1977; Chang et al., 1997; reviews by Xie and Carton, 2004; Chang et al., 2006a), the Benguela Niño (Shannon et al., 1986; review by Oettli et al., 2021), the Ningaloo Niño (Feng et al., 2013; review by Tozuka et al., 2021) and the North Pacific Meridional Mode (NPMM; Chiang and Vimont, 2004; review by Amaya, 2019), to name a few. Increasingly, the question arose to what extent variability in those remote regions was independent of ENSO, and whether it could influence the evolution of ENSO (see Chang et al., 2006a for a review, and Fig. 1 for a schematic). Thus, there was a growing interest in how the tropical oceans interact, and how these interactions may contribute to improved seasonal predictions of oceanic variability patterns and their impacts over land (Keenlyside et al. 2019).

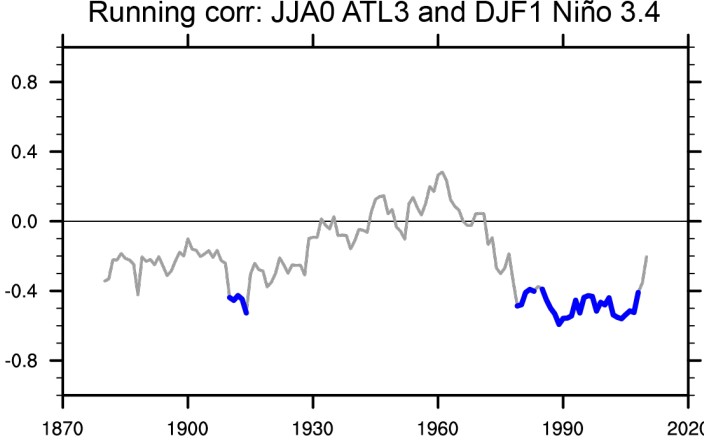

**Figure 2.   Running correlation of the June-July-August (JJA) ATL3 SST and the following December-January-February (DJF) Niño 3.4 SST using a 21-year sliding window for the period 1870-2021. The SST is from the CMIP6 amip experiment (see section 3). Correlation significant at the 95% level is indicated by the thick blue line segments. The significance test evaluates the null-hypothesis that the correlations are due to chance, using bootstrapping with 10,000 samples generated by randomly reshuffling 1-year blocks (Wilks, 1997). The figure suggests a strengthening of the equatorial Atlantic influence on ENSO since the 1970s, as suggested by Rodriguez-Fonseca et al. (2009), and a potential weakening at the end of the analysis period. Some of the experiments proposed for TBIMIP can address the potential dependence of such modulations on changes in background state, SST anomaly patterns, and radiative forcing.**

In addition to interannual variability patterns, such as ENSO, AZM, and IOD, there are also decadal and multi-decadal variability patterns, both in the tropics (e.g., the Tropical Pacific Decadal Variability (TPDV); see Power et al., 2021; and Capotondi et al., 2023, for a review; and the decadal IOD as reported in Ashok et al., 2004, and reviewed by Han et al., 2014) and in the extratropics (e.g., the Pacific Decadal Oscillation (PDO; Zhang et al., 1997; Mantua and Hare, 2002; review by Newman et al., 2016) and the Atlantic Multidecadal Variability (AMV; Kushnir, 1994; reviews by Keenlyside et al., 2015 and Zhang et al., 2019)). Due to their long timescales and extratropical locations, the latter patterns may influence other basins

through different pathways (e.g., Ruprich-Robert et al., 2017). The associated background changes may also modulate the way ocean basins interact on shorter timescales (Yu et al., 2015; Martin-Rey et al., 2015; Kajtar et al., 2018; McGregor et al., 2018; Drouard and Cassou, 2019). In addition, suppressing tropical basin interaction (TBI) in numerical experiments has been found to shift ENSO variability to lower frequencies (e.g., Kajtar et al., 2017; Kido et al., 2022, Bi et al., 2022, Zhao and Capotondi, 2024). It should also be noted that some of the interannual variability patterns of interest have considerable variance at decadal time scales. These include the central Pacific El Niño (Sullivan et al., 2016), and the AMM (e.g., Chang et al., 2006a). Thus, the decadal and longer timescales are of interest to the study of TBI but the observational record is short when low-frequency variability is the focus. The limited sample size of decadal-scale events, such as the AMV, as well as the existence of dedicated sensitivity experiments in the Coupled Model Intercomparison Project phase 6 (CMIP6) Decadal Climate Prediction Project (DCPP; Boer et al., 2016) have motivated us to focus the proposed experiments on interannual timescales while still considering the role of decadal modulation of remote influences, e.g., that of the equatorial Atlantic on ENSO (Fig. 2).

To study TBI, observational analysis is an obvious tool. Unfortunately, the observational record is relatively short, as mentioned above, with about 60-70 years of reliable data. For ENSO, e.g., this translates into roughly 20-30 events, and even less if only major events are considered. Given the considerable event-to-event diversity of ENSO (e.g., Timmermann et al., 2018), it is clear that the length of the observational record is a serious limitation when addressing interbasin interaction, particularly for statistical analysis and causality assessments. The event-to-event diversity further increases when considering the variability patterns in all three tropical ocean basins. A La Niña event, e.g., may be accompanied by a positive AZM event in one year, by a negative IOD in another, and by a combination of positive AMM and positive IOD in yet another. Thus, every year in the observational record features its own unique constellation of variability patterns in the three ocean basins, rendering the seemingly long 70-year observational record insufficient for disentangling the complex interactions. This is only complicated by the long-term changes in radiative forcing during the observation period.

Paleo proxies can substantially extend the data record available for analysis and have been used in the study of TBI (e.g., Cobb et al., 2001; Leduc et al., 2007). Proxy data, such as water isotopes ratio, however, must be converted into the variables of interest using a number of assumptions, which can contribute to uncertainties. Furthermore, the temporal resolution of such records may not always be high enough to resolve the variability patterns of interest, particularly when data for a particular season are desired. There is also uncertainty associated with the dating of proxies. Finally, the spatial coverage is sparse, particularly in the tropical Atlantic.

Climate model experiments offer several advantages, such as long simulations (1000 years or more) under steady radiative forcing, as is the case for the pre-industrial control simulations of CMIP6 (Eyring et al., 2016). In addition, climate model simulations allow experimentation, such as prescribing sea surface temperatures (SSTs) in one basin and analyzing the response in other basins. This avenue of investigation has been pursued by many groups, and numerous papers have been published (see Cai et al., 2019, for a review). Some of these studies, however, have arrived at diverging results. There is, e.g., disagreement on the role of the tropical Atlantic in influencing ENSO evolution, as illustrated by the composite of positive AMM events (Fig. 3), based on SST from the ERA5 reanalysis (Hersbach et al. 2018). Some studies argue for a strong influence

(e.g., Rodriguez-Fonseca et al., 2009; Ding et al., 2012; Ham et al., 2013ab; Martin-Rey et al., 2015), others for a limited
influence (Exarchou et al., 2021; Richter et al., 2021; Richter et al., 2023; Zhao and Capotondi, 2024), while yet some other
studies dismiss this influence as a statistical artifact (Zhang et al., 2021; Jiang et al., 2023). Both the atmosphere and ocean
allow for interaction pathways through material flow and waves, and these pathways have no built-in directionality, i.e., if the
Pacific can influence the Atlantic then the Atlantic can influence the Pacific. However, given the size of the Pacific basin and
the amplitude of ENSO, it is valid to question the importance of outside influences on ENSO. This is one of the motivations
for the TBI experiments described here.

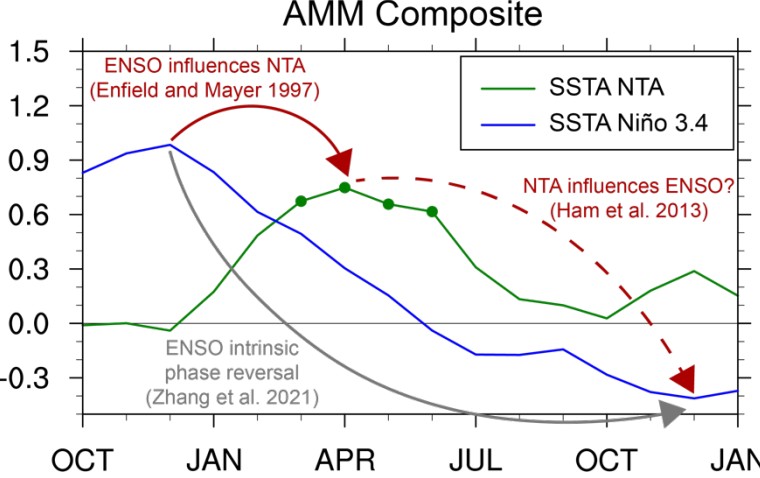


**Figure 3. ERA5 SST anomalies in the northern tropical Atlantic (NTA; 40-10W, 10-20N; green line) and Niño 3.4 (blue line) regions,**
**composited on positive AMM events, which are defined here as SST anomalies in the NTA region exceeding 0.8 standard deviations**
**in March-April-May. The years 1979, 1980, 1981, 1983, 1987, 1988, 1997, 1998, 2005, and 2010 are selected by this criterion. Values**
**significant at the 95% level are marked by dots (note that none of the Niño 3.4 values are significant). The composite shows that**
**NTA events tend to be preceded by El Niño events, a well-known remote impact of ENSO (indicated by the curved red arrow;**
**Enfield and Mayer, 1997). Furthermore, there are weak La Niña conditions in the winter following the peak of the positive AMM**
**event. This has been interpreted as the NTA influencing ENSO (dashed curved red arrow; Ham et al., 2013), but some studies have**
**challenged this, including Zhang et al. (2021), who suggest that the apparent influence stems from a misinterpretation of ENSO's**
**intrinsic phase reversal (i.e., El Niño events tend to be followed by La Niña, regardless of any tropical Atlantic SST anomalies;**
**curved grey arrow). The experiments proposed for TBIMIP will allow evaluating the importance of the NTA influence on ENSO.**
There is also an enduring conundrum as to why the strong ENSO signal in boreal winter has a robust influence on the northern
tropical Atlantic in spring (Enfield and Mayer, 1997) but an inconsistent influence on the adjacent equatorial Atlantic in
summer (Chang et al., 2006b; Lübbecke and McPhaden, 2012). While some robust impacts on the equatorial Atlantic have
been identified (Tokinaga et al., 2019; Jiang et al., 2023; Richter et al., 2024), it is still not fully understood why the major
1982-83 and 1997-98 El Niños were followed by negative and positive AZM events, respectively (Fig. 4). Finally, the
relationship between ENSO and the IOD has been probed in various climate model experiments, and these have arrived at
conflicting results, with some arguing for an IOD that is mostly independent of ENSO (e.g., Behera et al., 2006) or one that
may even influence ENSO (Behera and Yamagata, 2003; Luo et al., 2010), and others for an IOD that is largely controlled by
ENSO (e.g., Stuecker et al., 2017a). Recent work has also indicated that different flavours of the Indian Ocean Basin mode
can alter the decay of El Niño events (Wu et al., 2024).

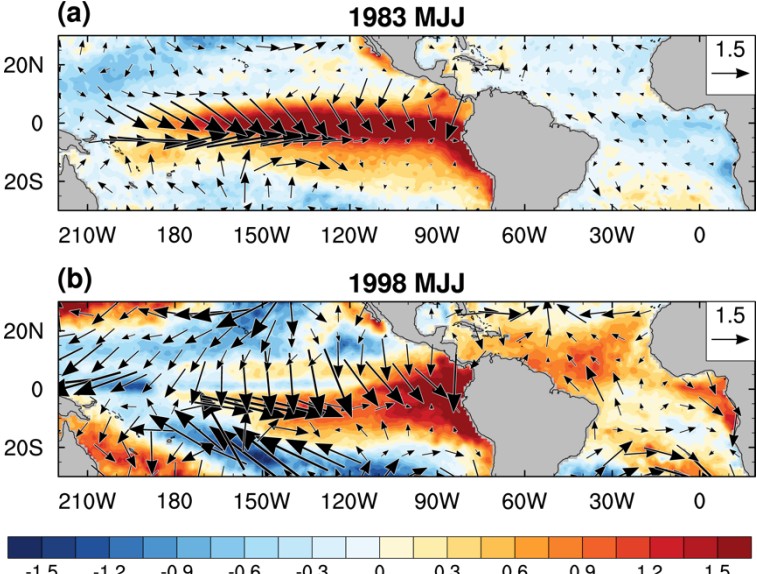


**Figure 4. Anomalous SST (shading; degC) and 10m winds (vectors; reference 1.5 m/s) averaged over May-June-July (MJJ) for (a)**
**1983 and (b) 1998. The fields are from the ERA5 reanalysis (Hersbach et al., 2018; note that SST is not an assimilated variable but**
**a blend of various observational products). The remnants of the very strong 1982-83 and 1997-98 El Niño events are evident in the**
**warm tropical Pacific SST anomalies. In the equatorial Atlantic, in contrast, SST anomalies are of the opposite sign during those**
**two years.**

There are at least two reasons why different models may provide conflicting results. One is that experiments by different
groups follow different protocols. This may include the way that perturbations are implemented in the model code but also
different simulation and analysis periods. The other is that systematic model errors (e.g., due to the use of different convective
parameterizations), substantially influence the outcome of such experiments. Since such errors differ widely across models,
the outcome of two sensitivity experiments conducted with different models can yield conflicting results even if they follow
the same protocol.
The proposed experiments can be categorized as "pacemaker" experiments, in which the atmospheric surface heat flux is
modified to constrain the model SSTs to follow observations. Hereafter, we will refer to this simply as SST restoring. The
overarching goal of the pacemaker experiments proposed for TBIMIP is to gain a deeper understanding of TBI and its potential
role in seasonal-to-decadal predictions. This includes a better understanding of the pathways involved and their relative
importance. Much of the interest in TBI stems from its potential to increase the skill of seasonal predictions, particularly that
of ENSO and its global impacts. Quantifying the contribution of TBI to prediction skill is therefore one of the major goals of
the TBI experiments, and a subset of the experiments is dedicated to this goal.

## 2 Justification for the TBI Model Intercomparison Project (TBIMIP)


While many experiments have been performed to explore TBI, these have followed varying experimental protocols, which
makes it difficult to compare results, as discussed in section 1. This was one of the major motivations for proposing an
intercomparison project in which all models follow the same experimental protocol. Based on such coordinated experiments,
it will be possible to evaluate the model dependence and robustness of the pathways of TBI.
Many general circulation model (GCM) intercomparison projects have been conducted and their output is publicly available
in many archives, most notably those of CMIP, which are hosted by the Earth System Grid Federation (ESGF). This prompts
the question whether there is a need for yet another intercomparison study. We first note that while a wide range of
intercomparisons have been performed, none of them has been dedicated to TBI at interannual timescales. The DCPP
component of CMIP6 features some experiments that are related to TBIMIP. That project, however, focuses on decadal
variability, while TBIMIP focuses on interannual variability. Since the AMV is one of the most pronounced patterns on decadal
and longer time scales, most DCPP experiments are designed to examine AMV impacts. As such, they examine the impacts
of AMV-related SST anomalies, which evolve slowly and extend into the high latitudes. The only experiment that partially
overlaps with TBIMIP is the DCPP Tier 3 experiment "dcppC-pac-pacemaker", in which SSTs in the tropical Pacific are
restored to observations. In addition to only one model having performed this experiment, the DCPP's focus on decadal
timescales means that the settings are not ideally suited for exploring interannual TBI. The Global Monsoons Model Inter-
comparison Project (GMMIP; Zhou et al., 2016) also features one experiment that is related to TBIMIP. In hist-resIPO, SST
anomalies are restored to observations in the central and eastern tropical Pacific. Four models in the CMIP6 archive have
completed this experiment but the protocol differs from that of TBIMIP. Importantly, there are no corresponding experiments
for the tropical Atlantic and Indian Ocean. We thus believe that the TBIMIP experiments proposed here offer a unique
opportunity for exploring TBI and its role in climate variability. Due to its seasonal prediction component, TBIMIP will also
offer an up-to-date dataset for comparing the prediction skill of state-of-the art prediction systems.
While the proposed TBIMIP experiments are distinct from the DCPP experiments, they may provide complementary
information regarding the role of tropical processes in decadal climate variability. Further synergy with existing CMIP6
experiments is provided by the use of the existing CMIP6 experiment "historical" as the reference for one subset of the
proposed experiments, as explained in Section 3. This eliminates the need to run a separate control simulation, thereby reducing
TBIMIP's computational load. It also allows comparison with the numerous experiments that are derived from "historical" and
are available in the CMIP6 archive, such as the single forcing experiments in the Detection and Attribution Model
Intercomparison Project (DAMIP; Gillett et al., 2016).

## 3 Experiment design of TBIMIP

Here we describe the key details of the experiment design. The full description can be found at https://www.clivar.org/sites/default/files/documents/TBI_CoEx_design.pdf or https://doi.org/10.5281/zenodo.13864935. A summary of the Tier 1 and Tier 2 experiments is given in Table 1. Potential Tier 3 experiments are discussed in Appendix A1.

| | | branch 1: Standard pacemaker | | branch 2: Pacemaker hindcast | |
|---|---|---|---|---|---|
| | | Name | description | name | description |
| **Tier 1** | TBI-hist-ctrl | | Reference experiment: Coupled ocean-atmosphere simulation with radiative forcing from historical (up to 2014) and ssp585 (2015-2021). If historical has already been performed, only extension from 2015-2021 is needed. | TBI-hind-ctrl | Hindcast experiment for the period 1982-2021 with ocean initialization in February (mandatory), and May, August, November (recommended). Depending on the initialization method, there may be the need for a separate control experiment. See experiment design for details. |
| | TBI-pace-P-anom | | Pacemaker experiment with SST restoring in the tropical Pacific (15°S-15°N). The restoring target is the model SST climatology plus observed SST anomalies | TBI-hind-P-anom | Restore SST anomalies in the tropical Pacific to lead-time dependent model climatology plus observed anomalies during forecast period. |
| | TBI-pace-A-anom | | Like TBI-pace-P-anom but for the tropical Atlantic (10°S-10°N). | TBI-hind-A-anom | Like TBI-hind-P-anom but for the tropical Atlantic. |
| | TBI-pace-I-anom | | Like TBI-pace-P-anom but for the tropical Indian Ocean (15°S-15°N). | TBI-hind-I-anom | Like TBI-hind-P-anom but for the tropical Indian Ocean. |
| **Tier 2** | | | | TBI-hind-ctrl | As in Tier 1. |
| | TBI-pace-P | | Like TBI-pace-P-anom but restore to full-field SST observations. | TBI-hind-P | Like TBI-hind-P-anom but restore to full-field observations. |
| | TBI-pace-A | | Like TBI-pace-A-anom but restore full-field SST observations. | TBI-hind-A | Like TBI-hind-P but for the tropical Atlantic. |
| | TBI-pace-I | | Like TBI-pace-I-anom but restore to full-field SST observations. | TBI-hind-I | Like TBI-hind-P but for the tropical Indian Ocean. |

| Tier 3 | | *reserved for future experiments* | | *reserved for future experiments* |
|--------|---|-----------------------------------|---|-----------------------------------|

**Table 1. Overview of the TBIMIP experiments. The latitudes refer to the core restoring regions. These are tapered off poleward over 10° buffer zones.**

Asi in other MIPs, the experiments are grouped into three tiers, with Tier 1 having the highest priority. Experiments in this tier use the anomaly restoring technique, while experiments in Tier 2 use full-field restoring to observations. Tier 3 is currently left for future additional experiments that may be suggested by analysis of the Tier 1 and Tier 2 experiments. Several suggestions for such experiments are given in Appendix A1. Both Tier 1 and Tier 2 are divided into two sets, or branches, of experiments. The first branch consists of standard pacemaker experiments, which are continuous integrations over the historical period from 1982-2021 (starting from 1870 is recommended) with SST restoring in selected basins. The second branch consists of pacemaker hindcasts for the period 1982-2021. These are initialized seasonal predictions with SST restoring in selected basins. (We note that we use "hindcast" in the sense of "reforecast", i.e. seasonal prediction experiments that are initialized from past observations.) Examples of such experiments can be found in the literature (e.g., Keenlyside et al., 2013; Luo et al., 2017). Participating groups can choose to perform only one of the two branches or both. For a given branch, however, all experiments should be performed.

Since the Tier 1 experiments use anomaly restoring, the SST target has to be calculated as the model SST climatology plus observed SST anomalies. The base period for calculating both the climatology and the anomalies is 1982-2019. The model climatology must be calculated from the reference simulation, which is TBI-hist-ctrl for the standard pacemaker and TBI-hind-ctrl for the pacemaker hindcast. For Tier 2, in contrast, the target SST is taken directly as the full-field observations.

The standard pacemaker experiments (branch 1) use the CMIP6 historical experiment as their control simulation. Groups that did not participate in CMIP6 should follow the CMIP6 protocol to perform the equivalent of historical. The radiative forcing is available via the ESGF website at https://pcmdi.llnl.gov/CMIP6/Guide/modelers.html. Where a pre-industrial control simulation (e.g., piControl in CMIP6) exists, a random year from that simulation should be chosen to initialize the control simulations. The CMIP6 forcing for the historical experiment is only available until 2014. It is suggested to use the radiative forcing from the ssp585 experiment for the period 2015-2021. However, since the radiative forcing does not vary much across scenarios for the first few years, any of these scenarios will be acceptable (Bi et al., 2022).

Three pacemaker experiments are requested, one for each of the tropical Pacific, the tropical Atlantic, and the tropical Indian Ocean. In each of these experiments, SSTs are restored to the target SSTs in the restoring region (10°S-10°N for the Atlantic, and 15°S-15°N for the Pacific and Indian Ocean; see section 4.3 for a justification of the narrower restoring region in the Atlantic). The restoring is linearly tapered to zero over a 10° buffer zone to the north and south of the core restoring region. The restoring time scale should be 15 days over a 50 m deep layer. The target SST is based on the boundary conditions of the CMIP6 amip experiments but extended to December 2022 (Paul Durack, personal communication). The amip SST boundary conditions, in turn, are derived from the Hadley Centre Sea Ice and Sea Surface Temperature data set (HadISST; Rayner et al., 2003) from January 1870 through October 1981 and the NOAA Optimum Interpolation SST (OISST) version 2 (Reynolds et al., 2002) from November 1981 through December 2022.

Masking has to be used to limit the SST restoring to the target basin. The restoring regions, including tapering zones, are
illustrated in Fig. 5. The core integration period for the standard pacemaker experiments is 1982-2021, but starting from 1870
is recommended, to allow for more robust analysis. The experiments should be initialized from the control simulation (CMIP6
historical or equivalent) and use the same radiative forcing. A minimum of 10 ensemble members is recommended. The method
of generating perturbed ensemble members is left to the modelling groups. One simple method is to slightly perturb the initial
atmospheric temperatures.

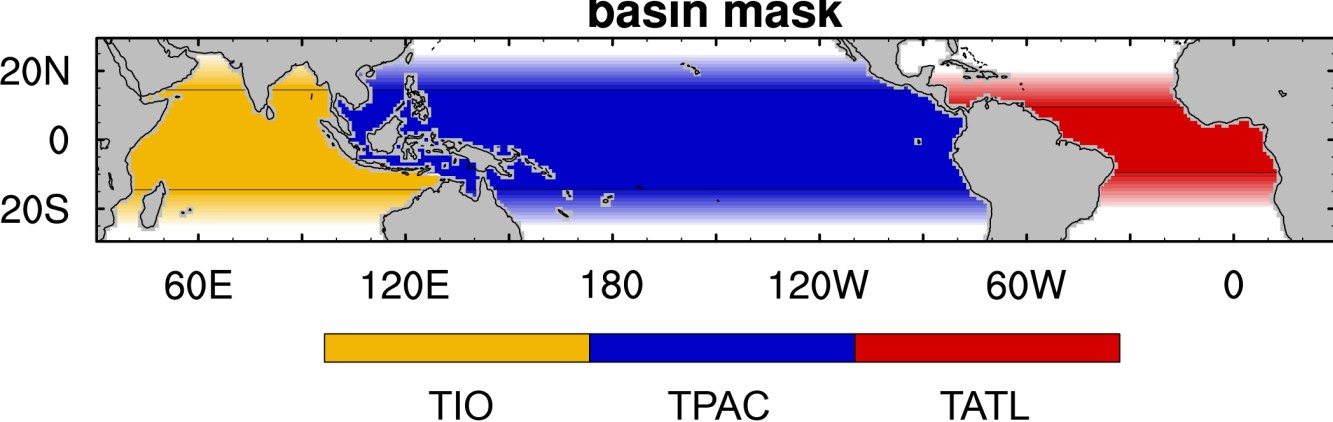

**Figure 5.** **The basin mask to be used for the TBIMIP experiments. See the section on "Data and code availability" for how to obtain**
**the data. The tropical Indian Ocean (TIO), the tropical Pacific (TPAC), and the tropical Atlantic (TATL) are indicated by yellow,**
**blue, and red shading, respectively. The core restoring regions are demarcated by horizontal lines, and the transition zones by**
**opacity gradients. Note the narrower meridional width of the tropical Atlantic restoring region.**
The pacemaker hindcasts (branch 2) are hindcast experiments with SST restoring in a selected basin. The control experiment
is a standard hindcast experiment. Many modelling groups may already have performed a hindcast experiment. Those who do
not must first complete this before performing the pacemaker hindcast experiments.
The technique for initializing the hindcasts (data assimilation etc.) is left to the modelling groups. While the initialization
method may influence the forecast skill and spread, it is not expected to strongly affect relative changes in the pacemaker
experiments, although future experiments should test this. The minimum requirement is one initialization on February 1 of
each year from 1982 through 2021. Each integration should be at least 12 months long. Additionally, initializations on May 1,
August 1, and November 1 are recommended. Finally, March 1 initializations may be useful for assessing prediction skill in
the equatorial Atlantic, due to the seasonality of the AZM.
Three pacemaker hindcast experiments are performed, one for each basin. The initialization method should be the same as for
the control hindcast. The restoring region and strength are the same as for the standard pacemaker experiments in branch 1.
The SST restoring starts with the initialization and is maintained throughout the forecast period. As for the standard pacemaker
experiments, a minimum of 10 ensemble members is recommended.

## 4 Climate model pacemaker experiments

### 4.1 Basic concept and rationale

At the heart of TBIMIP are coupled ocean-atmosphere experiments with SST restoring in selected target regions. Typically, the restoring target is a time-varying observed SST distribution, in which case the SSTs will follow the observations in the target region. In the wider sense of the meaning, pacemaker experiments can also restore to idealized SST distributions, such as a composite El Niño event, or a seasonal climatology. The general idea behind these pacemaker experiments is to examine the response of the atmospheric circulation and the subsequent impacts on the climate system outside the restoring region. A well-known example is the pacemaker experiment of Kosaka and Xie (2013), which examined how the global surface temperatures respond to prescribing SST in the central and eastern tropical Pacific. In particular, Kosaka and Xie (2013) were interested in how the tropical Pacific influences the evolution of the global temperature trend. Another example would be a pacemaker experiment in which SSTs are restored to observations in the tropical Atlantic in order to analyze the impacts of the tropical Atlantic on ENSO variability (e,g., Ding et al., 2012; Keenlyside et al., 2013; Exarchou et al., 2021; Liu et al., 2023). Such pacemaker experiments ask the question: To what extent will the climate system follow the observed evolution if one of its components is forced to follow observations? Tropical SSTs are an obvious candidate for this kind of intervention due to their strong influence on the atmospheric circulation. Other fields, however, can also be subjected to intervention, such as the surface wind fields (e.g., Richter et al., 2012; Ding et al., 2014; Gastineau et al., 2019; Voldoire et al., 2019), which have a strong impact on the ocean circulation and the surface enthalpy flux.

### 4.2 Methodology for SST restoring

There are several methods for constraining SST to follow a target time series. Below we list three potential methods but recommend using method 2).

1) Temperature nudging inside the ocean model

SST corresponds to the temperature of the uppermost vertical level of the ocean component. One approach is therefore to add a correction term to the temperature equation of the ocean model that nudges the SST toward the target value. The strength of the term is proportional to the difference between the target and model SST. This approach is akin to ocean data assimilation and has been employed in TBI studies (e.g., Ding et al., 2012; Chikamoto et al., 2016), and for the initialization of prediction experiments (Keenlyside et al., 2005; Keenlyside et al., 2013).

2) Surface heat flux term

The top ocean level interacts with the atmospheric model component through a coupler routine (e.g., Craig et al., 2017), which regulates the exchange of fluxes between the atmosphere and ocean. Another approach for modifying SSTs is therefore through manipulating inside the coupler routine the heat flux that goes into the ocean, which is the method recommended for the TBIMIP experiments. The heat flux in tropical regions consists of four components: net surface shortwave radiation, net

surface longwave radiation, latent heat flux, and sensible heat flux. Of these, the sensible heat flux is usually chosen for adding
the restoring flux (e.g., Kosaka and Xie, 2013).
3) Modifying SSTs "seen" by the atmospheric model
Because the flux coupler controls the SSTs that are "seen" by the atmospheric component, one can modify only this value,
thereby "tricking" the atmosphere into reacting to a temperature that is different from the actual ocean SST. This approach
leaves the ocean component completely unchanged (Richter and Doi, 2019). Furthermore, it allows the SSTs to exactly follow
a given distribution (as far as the atmosphere is concerned), rather than approximating it through correction terms. A potential
drawback is that this can lead to very unrealistic heat fluxes into the atmosphere (Wang et al., 2005).
Method 2) is recommended because it is commonly used, and because it allows SST restoring of variable strength, rather than
the prescribed SSTs of method 3). It should also be easier to implement than method 1), which requires modification of the
ocean model thermodynamic equation.
**4.3 Considerations when modifying the surface heat flux**
When constraining SSTs via the surface heat flux method, as recommended for the TBIMIP experiments, several issues need
to be considered.
First one has to decide on the strength of the restoring flux. The ocean mixed layer is an important concept to consider because
it is the layer that rapidly adjusts to the surface forcing. In the tropical oceans, a typical value for the mixed-layer depth (MLD)
is 50 m. Using this as a reference MLD, and based on the temperature difference between the actual and the target SST, one
can calculate the flux that is needed to achieve the target SST over a certain time scale:
$$F = (T_t - T_m)\rho C_p \frac{MLD}{\tau} \, , \tag{1}$$
where F is the correction heat flux [W m$^{-2}$], $T_t$ is the target SST [K], $T_m$ is the model SST [K], $\boldsymbol{\rho}$ is the density of seawater [kg
m$^{-3}$], $C_p$ is the heat capacity of seawater [J K-1 kg-1], MLD is the mixed-layer depth [m], and $\boldsymbol{\tau}$ [s] is the restoring time scale.
Thus, the heat flux needed increases with the deviation of the model SST from the target SST, the MLD, and the inverse of
the restoring time scale. It is clear from Eq. 1, that an instantaneous adjustment ($\boldsymbol{\tau}$=0, i.e., perfect agreement with the target
SST) would require an infinite heat flux. One therefore must compromise between the correspondence with the target SST and
a surface heat flux that is not overly disruptive. In the literature, a wide range of restoring time scales has been used. The
SINTEX-F1 seasonal prediction model (Luo et al., 2005) uses restoring time scales from 1 day to 3 days over 50 meters as a
simple data assimilation scheme for its forecasts. At the other end of the spectrum, restoring time scales of 30-60 days over 50
meters are used for decadal variability experiments, such as the CMIP6 DCPP. The IPSL decadal forecast system uses SST
nudging and a restoring time scale of 30 days as an assimilation scheme (Servonnat et al., 2015).
So, what are the reasons for not using short restoring time scales even though they allow for the highest correspondence with
the target SST? There are two main reasons. First, for short restoring time scales, the heat fluxes required can lead to very
unrealistic changes in the ocean circulation. Because the heat flux is absorbed in the top layer first, the immediate temperature
response could lead to unrealistic changes in vertical stability and, consequently, in vertical mixing. Second, overly strong
restoring can lead to unrealistic behaviour in regions where SST is primarily driven by the surface heat fluxes, rather than
driving them (Frankignoul, 1985; Frankignoul et al., 1998). This applies not only to extra-tropical regions but also to regions
of the Indian Ocean, Western Pacific, and North tropical Atlantic (Klein et al., 1999, Alexander et al., 2002, Wang et al., 2000).
In that case, strong restoring can affect the lead-lag relationship of SST and surface heat fluxes and even change the sign of
this relationship, as has been shown in the context of AMV pacemaker experiments. This, in turn, can lead to an inconsistent
large-scale response, when the SST-mediated changes in surface fluxes produce unrealistic diabatic atmospheric heating and
teleconnection patterns (Ding et al., 2014). In particular, some studies suggest that the role of the subtropical North Atlantic
may have been overestimated in experiments that performed SST restoring there (Kim et al., 2020; O'Reilly et al., 2023).

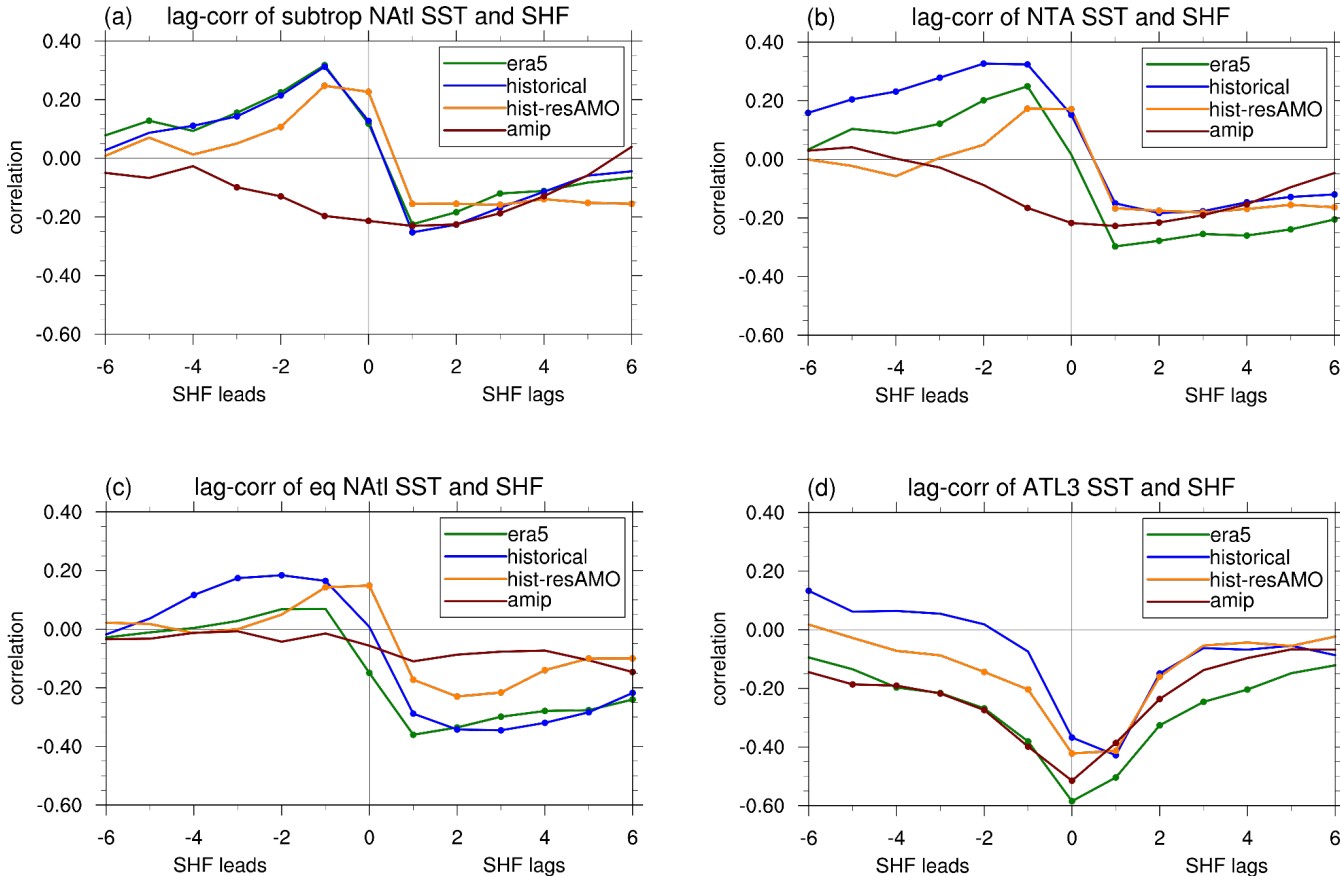


**Figure 6.** **Lead-lag correlation of anomalous SST and surface enthalpy flux (SHF; the sum of sensible and latent heat flux) for -6**
**to +6 months with SHF leading SST for negative lags. Positive correlations indicate that positive SST anomalies are associated with**
**SHF anomalies into the ocean. The data are from the ERA5 reanalysis (green line) and from the MRI-ESM2-0 CMIP6 model for**
**experiments historical (blue line), hist-resAMO (orange line), and amip (brown line). The analysis period is 1979-2014 for all**
**datasets. Filled circles indicate correlations that are significant at the 95% confidence level. The individual panels show the following**
**area averages: (a) subtropical North Atlantic (subtrop NAtl; 40-10W, 20-30N); (b) northern tropical Atlantic (NTA; 40-10W; 10-**
**20N); (c) equatorial North Atlantic (eq. NAtl; 40-10W; 5-10N); (d) ATL3 (20W-0; 3S-3N).**

Figure 6 examines the influence of SST restoring by examining the lead-lag relation between SST and surface enthalpy flux (SHF) for several regions that range from the subtropical North Atlantic (Fig. 6a) to the equatorial Atlantic (Fig. 6d; see figure caption for area definitions). The ERA5 reanalysis is compared to CMIP6 simulations with the MRI-ESM2-0 climate model from three different experiments: historical, with full ocean-atmosphere coupling; hist-resAMO (part of GMMIP), with relatively weak SST restoring (60 days over a 50 m layer) in the AMO region (core restoring region 5–65°N, 65–5°W, with 5° buffer zones in the meridional and zonal directions); and amip, with SST completely fixed. For both the reanalysis and the model simulations the analysis period is 1979-2014. In all three off-equatorial regions (Figs. 6a-c) the ERA5 reanalysis shows the highest positive correlation when SHF leads SST by one month, indicating that SHF anomalies are driving SST anomalies (Frankignoul et al., 1998). The lowest negative correlation occurs when SHF lags SST by one month, with low values for the contemporaneous correlation. This behavior is well reproduced by the MRI-ESM2-0 historical simulations and, to a somewhat lesser degree, by the hist-resAMO simulation, presumably due to the interference from the SST restoring. In the amip simulation, however, there are negative correlations for both SHF leading SST and SHF lagging SST, indicating that the model attempts to damp the SST anomalies at all times. This contrasts with both the reanalysis and the other model simulations and strongly suggests that the SST prescription disrupts the relation between SHF and SST.

In the equatorial Atlantic (Fig. 6d), conversely, there are no categorical differences across the four datasets, with both the reanalysis and the simulations showing negative correlations that are lowest around the contemporaneous correlation. This indicates that the ocean circulation drives SST anomalies, while the atmosphere damps them through SHF anomalies.

Given that SST restoring can lead to unrealistic fluxes outside the deep tropics, as suggested by Fig. 6, it is advisable to limit the meridional width of the restoring region. We therefore restrict the core restoring region from 10°S to 10°N in the tropical Atlantic, and from 15°S to 15°N in the tropical Pacific and Indian Ocean, with 10° transition zones in each hemisphere. The smaller meridional extent of the tropical Atlantic restoring region is motivated by the fact that deep convection is more confined around the equator there, and by studies indicating unrealistic fluxes in the subtropical North Atlantic when SSTs are restored there (Kim et al., 2020; O'Reilly et al., 2023).

An important choice to make is whether to use full-field or anomaly SST restoring. In full-field restoring, the target SST field is the total observed SST, i.e., observed SST climatology plus observed SST anomaly. In anomaly restoring, on the other hand, the target is model climatology plus observed SST anomaly. The advantage of anomaly restoring is that it preserves the model SST climatology in the restoring region, so that it remains consistent with the climatology outside the restoring region, thus reducing the effect of sharp gradients. In the equatorial and southern tropical Atlantic, models tend to have a pronounced warm bias (e.g., Richter and Tokinaga, 2020). Under such circumstances, prescribing the observed climatology will reduce the average SST in the region and may fundamentally change the way it interacts with other basins. Anomaly restoring therefore offers a way to avoid undesirable side effects of the SST intervention. A potential disadvantage for a multi-model intercomparison is that the total prescribed SST values will be different for every model. This may make it more difficult to compare results across models. In addition, the method requires some consideration on how to calculate the target SSTs. To illustrate this, we introduce a few equations. The total model SST can be written as the sum of a climatology and an anomaly:

$T_m = \bar{T}_m + T_m'$, where the overbar denotes the seasonally varying climatology, and the prime denotes the anomaly. Likewise,
the total observed SST can be written as $T_o = \bar{T}_o + T_o'$. For anomaly restoring, the restoring target is the sum of model
climatology and observed anomaly: $T_t = \bar{T}_m + T_o'$. An energy imbalance can occur in the model if there is a mismatch between
the restoring target and the model SST of the free-running control simulation: $E = T_t - T_m = \bar{T}_m + T_o' - (\underline{\bar{T}_m} + T_m') = T_o' -$
$T_m'$. If this imbalance accumulates over the integration period, it can potentially change the SST distribution outside the
restoring region and adversely affect the outcome of the pacemaker experiment. Such an imbalance can occur if the base period
(used for the calculation of the climatology) is different between model and observations, due to the warming trend during the
historical period. It is therefore important to use a consistent base period when calculating the restoring target. Even with the
same base period, however, an imbalance can occur if the base period is much shorter than the integration period. As an
example, consider a case where we define the base period as 1982-2019 but perform the pacemaker experiment over the period
1870-2021. Both the model and the observed SST anomalies are calculated relative to the same 1982-2019 base period: $T_m' =$
$T_m - \bar{T}_m^{(1982\to2019)}$ and $T_o' = T_o - \bar{T}_o^{(1982\to2019)}$, where, without loss of generality, we neglect the seasonal dependence of the
climatology. The time-integrated imbalance then becomes

$$\int_{t1}^{t2} E\,dt = \int_{t_1}^{t_2}(T_o' - T_m')dt = \int_{t_1}^{t_2}(T_o - T_m)dt - \int_{t_1}^{t_2}(\bar{T}_o^{(1982\to2019)} - \bar{T}_m^{(1982\to2019)})dt \qquad (2)$$

where $t_1$ and $t_2$ denote the integration period of the pacemaker experiment. Noting that the second term on the right-hand side
of equation (2) is constant, and dividing by the integration period, we obtain

$$\bar{E}^{t1\to t2} = \bar{T}_o^{(t1\to t2)} - \bar{T}_m^{(t1\to t2)} - [\bar{T}_o^{(1982\to2019)} - \bar{T}_m^{(1982\to2019)}] \qquad (3)$$

If the integration period is equal to the base period ($t_1$=1982, $t_2$=2019), the imbalance is identical to zero. Non-trivial imbalances
can arise when the integration period is substantially longer (e.g., 1870-2021, as in our example) and if the difference between
model and observed SST substantially changes over the longer period. In other words, problems arise when the simulated and
observed SST trends are substantially different. We test this for a few selected models participating in the CMIP6 historical
experiment (Fig. 7a), using as the observational reference the CMIP6 amip SST, which is derived from HadISST and OISST
(see section 3). The area average of SST over the tropical Pacific varies substantially across models, with the warmest model
being almost 1.5 degC warmer than the coldest model, and the observations roughly in the middle. This bias spread, however,
is of no concern for our experiments because the bias itself does not enter into the energy imbalance. The important question
is whether the gap between a given model and the observations varies substantially over time. We therefore remove the time
mean and replot the SST evolution (Fig. 7b). The curves are now more closely spaced, suggesting that the bias of a given
model does not vary substantially over time, although the well-known trend overestimation at the beginning of the 21st century
is evident (Kosaka and Xie, 2013; Wills et al., 2022; Beverly et al., 2024). We conclude that using a shorter base period should
not lead to major imbalances though this should be carefully evaluated for each model. Calculating the imbalance (term $E$ in
equation (3)) yields the values shown in Table 2, where unlike in Eq. (3), the shorter base period is 1977-2014 (rather than
1982-2019) because this is readily available in the CMIP6 historical simulations.

| Model | CanESM5 | CESM2 | CNRM-CM6-1 | EC-Earth3 | FGOALS-f3-L | GISS-E2-1-G | IPSL-CM6A-LR |
|---|---|---|---|---|---|---|---|
| Imbalance (K) [term E in eq. 3] | 0.24 | 0.04 | 0.01 | 0.12 | 0.10 | 0.03 | 0.15 |

Table 2. Imbalance (K) incurred by using a base period (1977-2014) that is much shorter than the integration period (1870-2014) when calculating the model climatology and observed anomalies (see Eq. (3) for an explanation) in historical simulations of seven CMIP6 models, as indicated in the top row.

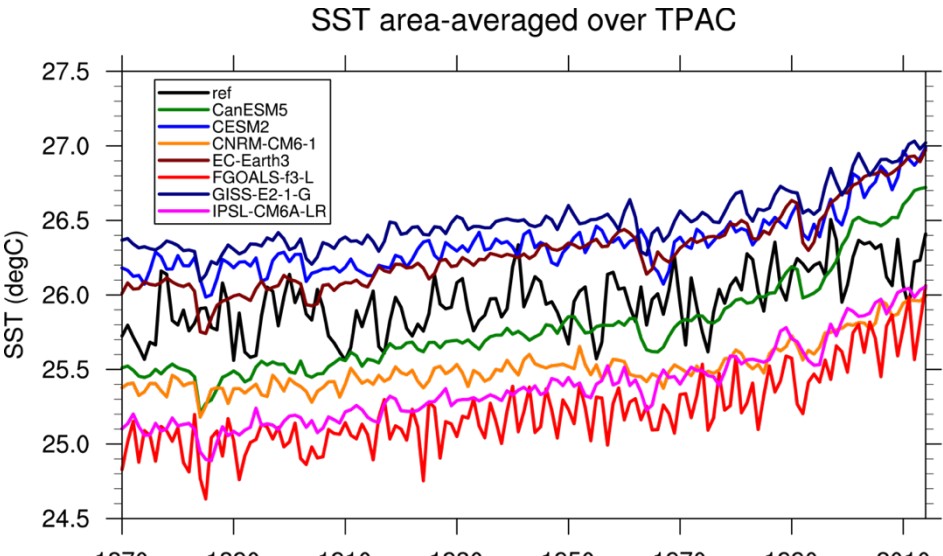

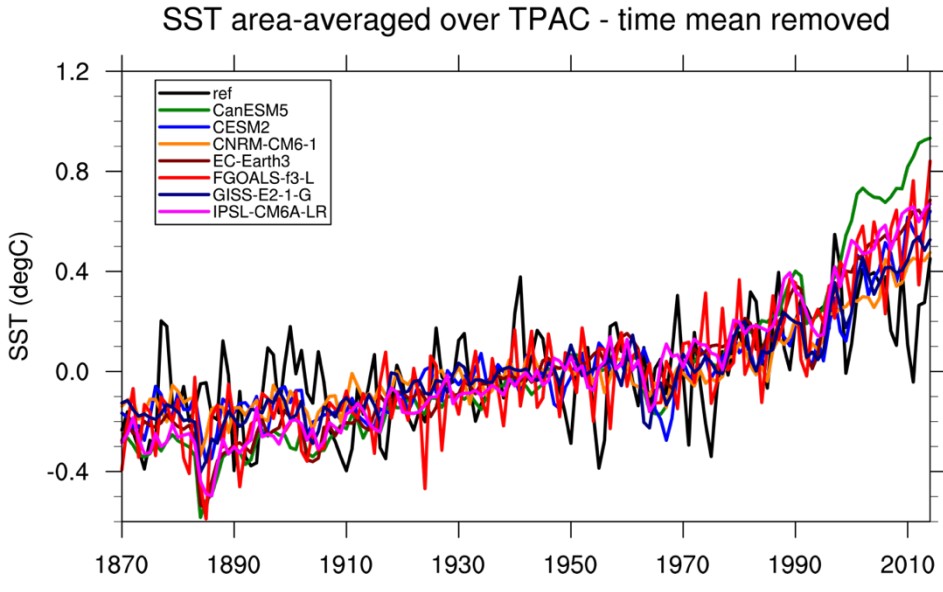

**Figure 7.** SST (C) averaged over the tropical Pacific (entire basin width, 30°S-30°N) for the reference (CMIP6 amip SST) and 7 models from the CMIP6 historical experiment as indicated by the legend in the upper left of each panel. For the models, the lines represent the average over all respective ensemble members. The panels show (a) full field SST, and (b) the deviation of the full-field SST from its 1870-2014 time average for each dataset.

Following the above analysis, we define 1982-2019 as our base period. Using this relatively short base period for TBIMIP is motivated by the fact that it is a subset of the minimum period requested for all TBIMIP simulations. Thus, this period should be available to all participating groups. In particular, the pacemaker hindcast experiments (see section 3) will only be performed for the period 1982-2021, meaning that a longer base period would not be possible for those experiments.

When restoring SSTs in a particular ocean basin, one has to consider not only the meridional but also the zonal extent of the restoring region. For the tropical Atlantic, the American and African coastlines provide an obvious choice for a basin mask. The boundary between the tropical Pacific and Indian Ocean is not as obvious because the Indonesian Throughflow is a porous boundary. Some previous experiments have avoided this problem by excluding the entire western tropical Pacific (e.g., Kosaka and Xie, 2013). For TBIMIP, we choose to extend the tropical Pacific region all the way to the Maritime Continent, according to the basin mask provided by the World Ocean Atlas (Locarnini et al., 2010). Some modifications were performed to simplify the basin mask (Fig. 5). This mask is publicly available. See the "Data and code availability" section for how to obtain the data.

## 4.4 Drawbacks of pacemaker experiments

While pacemaker experiments are a useful tool for understanding the interaction among the tropical basins, they also have potential shortcomings.

*1) The infinite heat source problem*

SST restoring can lead to a potentially infinite heat source or sink. The larger the difference between restoring target and model SST, the larger the heat flux that has to be pumped into the ocean or atmosphere (see Eq. 1). This adjustment flux is a purely mathematical entity and therefore not bounded by any energy constraints. In practice, this issue will be more prominent when full-field restoring is used and when there are large model biases. Even in anomaly restoring experiments, however, this issue can arise in regions where the atmosphere exerts an important influence on the ocean, such as in the subtropics. In such regions, the underlying assumption of SST pacemaker experiments that the SSTs drive the atmosphere is less valid, which can lead to unrealistic results, as discussed in section 4.3.

*2) Shift in the model dynamics*

The intervention in the model dynamics may perturb the simulation to such an extent that it fundamentally alters the basic flow. In that case, interpretation of the results may be difficult. Again, this factor will be more important when full-field restoring is used.

*3) Insufficient model fidelity*

If the simulated variability patterns are substantially different from those observed, it may be difficult to draw conclusions about nature. An example would be the seasonal preference of variability patterns. ENSO, e.g., is known to have its peak in

boreal winter and models are known to struggle with reproducing this seasonal synchronization (Stein et al. 2014; Liao et al. 2021). If a model ENSO peaks in summer, e.g., this may have serious repercussions on how it interacts with other basins. One of the reasons for TBIMIP is exactly to study this model dependence.

*4) Incomplete decoupling of basins*

While the goal of TBIMIP is to study the influence of individual basins on the climate system, this separation into individual basins cannot be completely successful. The SSTs one prescribes in the tropical Atlantic or Indian Ocean, e.g., implicitly contain some impact from the tropical Pacific because ENSO has contributed to shaping them. It is therefore not possible to completely isolate the influences of individual basins, and this should be borne in mind when analyzing the output from pacemaker experiments. When assessing impacts on predictability, for instance, it has been shown that experiments with relaxation toward observations greatly overestimate ENSO forecast skill because of the built-in presumed perfect evolution of the stochastic noise-driven component of SSTs as well as the aforementioned ENSO effect on remote SSTs (see discussion in Zhao et al., 2024).

*5) Reliability of the observations*

In addition to 1) – 4), which are limitations inherent to the modelling approach, there is also the problem of the reliability of the observations used to design the restoring target. This is mainly an issue for the pre-satellite era, when SST measurements mostly relied on shipboard observations. Thus, this issue can potentially affect the pacemaker experiments, if they are extended beyond the satellite observation period. Results from this period will have to be treated with caution.

Despite the caveats listed above, we do believe that pacemaker experiments are a valuable tool for gaining a deeper understanding of TBI.

## 5 Participation

The participation of multiple modelling groups is essential for the success of any MIP. At the time of writing, several groups have performed part of the experiments or are at the preparation stage, as detailed in Table 3. The participation of additional groups is highly welcome. The minimum requirement is the completion of at least one branch (standard pacemaker or pacemaker hindcast) of the Tier 1 or Tier 2 experiments. For the standard pacemaker branch, this consists of the control historical experiment and one experiment for SST restoring in each tropical basin. The minimum integration period is 1982-2021. Assuming 10 ensemble members, the minimum simulation time is 4 experiments x 10 ensemble members x 40 years per simulation, which equals 1600 simulation years. This reduces to 1200 simulation years if a historical simulation is already available.

| Model | Center | Type of experiment | Status |
|-------|--------|--------------------|--------|
| CESM2 | US NSF NCAR | hindcast+standard | completed |
| CESM2 | SCSIO, China | Tier 2 expmnts | completed |

| NorCPM | U. of Bergen | hindcast+standard | completed |
|---|---|---|---|
| SINTEX-F2 | JAMSTEC | pmaker hindcast | completed |
| MIROC6 | JAMSTEC, University of Tokyo, NIES | hindcast+standard<br>Tier 2 standard pmaker | completed |
| ACCESS-CM2 | CSIRO, Australia | standard pmaker | in preparation |
| IPSL-CM6A-LR | IPSL, France | standard pmaker | completed |

**Table 3.** **Status of the TBIMIP experiments execution as of February 2025. Unless explicitly noted, the status refers to Tier 1 experiments. "pmaker hindcast" and "hindcast" stand for the pacemaker hindcast branch, and "standard pmaker" and "standard" stand for the standard pacemaker branch of the experiments (see section 3).**

For the pacemaker hindcast experiments, the minimum requirement is one control hindcast experiment, and one SST intervention experiment for each basin. The minimum hindcast period is 1982-2021, with at least one initialization per year (on February 1) that is integrated for 12 months into the future. Thus, the minimum simulation time is 4 experiments x 10 ensemble members x 1 forecast initialization per year x 1 year per forecast x 40 years, which again equals 1600 simulation years.

The output variables that should be archived are listed in Table 4. They are grouped into three levels, with level 1 being the minimum requirement, level 2 desirable, and level 3 optional. The variable names follow the CMIP nomenclature, which can be found here: https://clipc-services.ceda.ac.uk/dreq/mipVars.html. All variables need to follow the CMIP conventions, including variable name and output format ("cmorization"). Vertical pressure levels for 3D atmospheric variables should follow the standard CMIP format (hPa): 1000, 925, 850, 700, 600, 500, 400, 300, 250, 200, 150, 100, 70, 50, 30, 20, 10, 5, 1, with a reduced number of levels for daily data, as indicated in Table 4.

One variable that is only found in a few of the CMIP6 experiments is *hfcorr*, which is the heat flux term applied to restore SST to the target value. This is an important diagnostic for examining the potential energy imbalance created by the heat flux correction and is also a measure for how much the ocean SST would diverge from the target SST if left unperturbed, i.e., the degree to which the ocean-atmosphere system resists the SST restoring. In many models, outputting this variable will require code modifications. Note that this variable should be separate from the sensible heat flux or latent heat flux variables, even though it may eventually be added to one of these in the flux coupler.

|  | 2D atmosphere | 3D atmosphere | 2D ocean | 3D ocean |
|---|---|---|---|---|
| **Level 1** | ts, uas, vas, pr, ps, psl, hfls, hfss, rsus, rsds, rlus, rlds, rlut, rsdt, rsut, tauu, tauv, cld, tas, sfcWind, hfcorr* | ta, ua va, wap, zg, hus | zos, tos, hfcorr, z20* (depth of the 20C-isotherm) | thetao |
| **Level 2** | daily mean: ts, uas, vas, pr, ps, ua200, va200, wap500 |  | uos, vos, mlotst, tauuo, tauvo, hcont300 daily mean: zos, uos, vos, z20 | uo, vo, wo, so |
| **Level 3** | mrso, prw, huss, hurs, sic, snd; daily mean: ta, ua, va, wap, zg, hus (reduced levels: 850, 500, 200, 100, 50 hPa) | cl, tntmp* (diabatic heating); components of tntmp* (latent, sensible, shortwave, longwave) | msftbarot, msftmz, hfbasin; daily mean: sos; ocean heat budget terms* | rhopoto ocean heat budget terms* |

**Table 4. Minimum requirements for output variables of the TBIMIP experiments in all three tiers and for both branches. The**
**CMIP vocabulary for variable names is used. Variables that may not be included in the standard output of models are marked by**
**an asterisk. If not indicated otherwise, monthly means are requested.**
We are aiming to make the model output available to the community through the CMIP6Plus project ([https://wcrp-](https://wcrp-cmip.org/cmip6plus/)
[cmip.org/cmip6plus/](https://wcrp-cmip.org/cmip6plus/)), which has been set up to bridge the interim period between CMIP6 and CMIP7. There will be an
embargo period during which data will be available only to participating groups and members of the Climate and Ocean -
Variability, Predictability, and Change (CLIVAR) TBI Research Focus. During this period, we will perform a quality check
of the data and perform some initial analysis. After the embargo is lifted, the data will be made available to the community,
just as other CMIP6 data. Under the current timeline, this is anticipated to happen in mid-2025.
**6 Discussion of complementary approaches to investigating TBI**
The experiments of TBIMIP were conceptualized by the CLIVAR Research Focus on Tropical Basin Interaction. These
experiments are useful for probing the interaction among the tropical ocean basins but also have their limitations, as discussed
in Section 4.4. TBIMIP should therefore be viewed as one tool for understanding TBI, rather than delivering a definitive
answer. Indeed, the CLIVAR Research Focus on Tropical Basin Interaction is involved in a range of activities aimed at
fostering observational and paleo proxy research, as well as the use of conceptual models and statistical analysis. Below, we
therefore discuss additional approaches to complement the output from TBIMIP, with the aim of highlighting ongoing research
efforts and encouraging future experimentation and analysis.
Held (2005) advocated for the use of a hierarchy of models to advance understanding of the climate system, with models
ranging from conceptual to highly complex. Subsequent studies have elaborated on this concept (e.g., Jeevanjee et al., 2017;
Stuecker, 2023). The recharge oscillator (Jin, 1997) can be considered as a prime example of a conceptual model and is one of
the simplest models capable of reproducing observed ENSO behaviour. Initially designed for the tropical Pacific only, this
model has been extended to include interactions with other regions (Jansen et al., 2009). Most recently, Zhao et al. (2024) have

presented an extended recharge oscillator with remarkable ENSO prediction skill. This model is being made available to the community and should be a useful tool for studying TBI. Its low complexity will facilitate the interpretation of experimental results.

Another simple approach for modelling the climate system is the linear inverse model (LIM; Hasselmann, 1988; Penland and Magorian, 1993). While typically somewhat more complex and less amenable to intuitive physical understanding than the recharge oscillator, LIMs offer a rich framework of analysis tools, such as optimal precursors (Penland and Sardeshmukh, 1995) and principal oscillation patterns (Hasselmann, 1988; von Storch et al., 1995). Recently, LIMs have been modified to allow for the study of TBI (Zhang et al., 2021; Alexander et al., 2022; Kido et al., 2022; Jin et al., 2023; Zhao et al., 2023; Zhao and Capotondi, 2024). The technique involves splitting the LIM operator matrix into submatrices that represent the interaction between two basins and then selectively setting those submatrices to zero. The interbasin LIM developed by Kido et al. (2022) will be made available to the community.

Intermediate complexity models (ICMs) are situated halfway between conceptual models and GCMs. The Cane-Zebiak (CZ) model (Cane and Zebiak, 1987) consists of a reduced-gravity ocean and a shallow-water-equation atmosphere component, the latter based on the work by Gill (1980). While originally developed for the tropical Pacific to study and predict ENSO, it has also been adapted for the tropical Atlantic (Zebiak, 1993). A CZ model for the interaction between the three tropical ocean basins could be an important addition for the study of TBI, as it could bridge the gap between conceptual models and GCM experiments.

Another example of an ICM is the SPEEDY model, developed by Molteni (2003). The code of this model is available to the community and has been used by a number of researchers to study TBI (e.g., Sun et al., 2017; Molteni et al., 2024). The SPEEDY model can be used as a stand-alone AGCM, or can be coupled to either a slab ocean model (Molteni et al., 2024) or a full complexity ocean model (Ruggieri et al., 2024). The advantage of this type of model is that the atmospheric component is very fast compared to state-of-the-art climate models, allowing to perform more than 100 years of simulation in 24 hours on a single CPU, while reproducing observed large-scale climate variability similar to state-of-the-art models. This computational efficiency advantage remains even when coupled to complex ocean models (Kucharski et al., 2016a,b). Indeed, in Kucharski et al. (2016b), several previously proposed ways of Tropical Atlantic mode forcing of Pacific climate variability have been revisited from interannual to multidecadal time-scales in ensembles of century-long pacemaker experiments. The relative simplicity of the model code allows modifications that may be used to efficiently test hypotheses for TBI.

Toward the complex end of the spectrum, GCM experiments with idealized boundary conditions, such as simplified geometries or SST patterns, or idealized narrowband forcing timescales (e.g., Su et al., 2005; Stuecker et al., 2015; Stuecker et al., 2017a,b; Stuecker, 2018), may offer a way to increase our understanding of TBI. Recently, Dommenget and Hutchinson (2025) have performed TBI experiments with idealized land-sea configurations. A twin Pacific configuration, for instance, highlighted clearly how tropical basin interaction can lead to synchronized and highly amplified variability in the tropical oceans. This concept helps to understand how tropical basin interaction develops in simplified setups, and how it transforms into more complex, less obvious interaction in more realistic setups. The output from these experiments will be made available to the

community. Another form of idealized GCM experiments consists of restoring SSTs to climatology in a specified region, which allows exploring how the absence of certain variability patterns, such as ENSO, influences the atmospheric circulation (Richter and Doi, 2019) and remote basins (Kataoka et al., 2018; Liguori et al., 2022).

Machine learning (ML), in particular deep learning, is increasingly being used for predicting interannual climate variability (e.g., Ham et al., 2019; Zhou and Zhang, 2023). While ML is often seen as the epitome of a black box approach, impervious to human understanding, there are efforts to remedy this problem (e.g., Gibson et al., 2021; Bommer et al., 2024), such as identifying predictors (Shin et al., 2022) or using ML to discover prediction equations via symbolic regression (Brunton et al., 2016; Najar et al., 2023). Such approaches may also be useful for the study of interbasin interaction, by identifying key regions and pathways influencing another basin, or by devising simple models of TBI.

In addition to deep learning, there are other nonlinear statistical approaches. One of them is complex network analysis, which has been applied to various TBI-related topics, such as identifying teleconnections of the Indian summer monsoon (Di Capua et al., 2020), and the linkage between the tropical Atlantic and Pacific (Karmouche et al., 2023). Other methods that can be brought to bear on TBI include generalized event synchronization analysis (Mao et al., 2022), and analog-models (Ding and Alexander, 2023).

Common to all the conceptual models and statistical methods described above is that they are, to a large extent, data driven. Some conceptual models like the recharge oscillator may be devised using physical understanding but eventually require fitting their parameters to observations, because these cannot be derived from first principles. Thus, all these models require training and validation on the limited observational data record (see discussion on the length of the available data record in section 1). The number of adjustable parameters is quite limited for conceptual models like the recharge oscillator but rapidly grows with the complexity of the model, with deep learning known to be data-intensive. This may be another obstacle standing in the way of ML being applied to climate sciences and the study of TBI. While the observational record is short and confounded by changing radiative forcing, long climate simulations under steady radiative forcing are available. These climate simulations are subject to systematic errors, as discussed in section 1, and therefore training data-driven models on GCMs may have its limitations. On the other hand, ML and conceptual models trained on GCM output may help to understand the behaviour of GCMs and the way they portray TBI. Thus, tools like the recharge oscillator, LIMs, and/or ML models could be used to augment the results of GCM experiments.

We conclude that many tools are available for analysing TBI, all with their own strengths and weaknesses. Optimally combining these tools is a difficult task but crucial for gaining a deeper understanding of TBI. Fostering the development of such tools and their application to TBI is one of the priorities of the CLIVAR Research Focus on Tropical Basin Interaction. We hope that the coordinated GCM experiments will be one useful contribution toward this goal.

## 7 Summary

Interaction among the tropical basins is a crucial component of the climate system. A deeper understanding of TBI holds the key to improved predictions of subseasonal to decadal climate variability, and to projecting how this variability will change under greenhouse gas forcing. The TBIMIP introduced here, aims to make progress in this direction through a set of multi-model coordinated GCM experiments. As shown in section 6, there are alternative and complementary approaches using conceptual models and statistical approaches. The strength of GCM experiments lies in their comprehensive depiction of the climate system, which allows analyzing the physical mechanisms of TBI, thus contributing to our understanding of TBI. Furthermore, GCMs are primarily based on fundamental physical laws and thus, unlike data-driven models, are not limited by the relatively short observational data record. While GCMs are subject to biases, the multi-model approach will allow assessing the influence of these model biases on the model results. In addition to offering a rich dataset for the analysis of TBI and its underlying mechanisms, TBIMIP will also allow us to quantify the importance of individual pathways. This should contribute to a deeper understanding of TBI and to reconciling conflicting results of previous studies. By making the datasets from the experiments available to the community we hope to stimulate research in this important research area.

*Data and code availability.* The ERA5 reanalysis data were obtained from https://www.ecmwf.int/en/forecasts/datasets/reanalysis-datasets/era5. ETOPO5 was obtained from the National Geophysical Data Center, NOAA, at https://doi.org/10.7289/V5D798BF. The CMIP6 model datasets are available from the Earth System Grid Federation (ESGF) at https://esgf- node.llnl.gov/search/cmip6/. The amip SST boundary conditions are available from the ESGF website at https://aims2.llnl.gov/search/input4mips/, by setting "MIP Era" to CMIP6Plus and variable name to tosbcs, version 1.1.9. The HadISST and OISST, on which the amip SST are based, can be obtained from https://www.metoffice.gov.uk/hadobs/hadisst/data/download.html and https://psl.noaa.gov/data/gridded/data.noaa.oisst.v2.html, respectively. The basin mask used to create Fig. 5 can be found at https://doi.org/10.5281/zenodo.13865022. Note that the meridional restoring width to be used in the TBIMIP experiments is not indicated in this data set.

The code to produce the figures can be found at https://zenodo.org/records/14000123.

*Author contributions.* IR prepared the manuscript and drafted the figures with contributions from all authors.

*Competing interests.* The authors declare no competing interests.

*Acknowledgements.* The authors would like to thank Michael Alexander, Antonietta Capotondi and one anonymous reviewer for their constructive comments and suggestions. IR was supported by the Japan Society for the Promotion of Science through Grant-in-Aid for Scientific Research (KAKENHI), Grant JP23K25946, and the Kyushu University Program for Collaborative Research, Grant 2024CR-AO-2. MFS was supported by NSF grant AGS-2141728. This is IPRC publication X and SOEST contribution Y. AH was supported by the Regional and Global Model Analysis (RGMA) component of the Earth and Environmental System Modeling Program of the U.S. Department of Energy's Office of Biological & Environmental Research (BER) via the US National Science Foundation (NSF) IA 1947282 (DE-SC0022070). The US NSF National Center for Atmospheric Research (NCAR) is sponsored by the US NSF under Cooperative Agreement No. 1852977. YMO is supported by NSF grant AGS-2105641. PC is supported by NSF grant AGS-2231237, and the NOAA grant NA20OAR4310408 and NA24OARX431G0047-T1-01. SY is supported by NOAA grant NA20OAR4310408. CW was supported by the National Natural Science Foundation of China (W2441014 and 42192564). TK was supported by MEXT program for the advanced studies of climate change projection (SENTAN) Grant Number JPMXD0722680395. NK and PGC were supported by the research council of Norway (#328935; #309562) and by Norwegian national computing and storage resources provided by UNINETT Sigma2 AS (NN9039K, NS9039K). WP was supported by the Korea Institute of Marine Science & Technology Promotion (KIMST) funded by the Ministry of Oceans and Fisheries, Korea (20220548, PM63990). AST thanks Zoe Gillett, Paola Petrelli, and the ARC Centre of Excellence for Climate Extremes for the pace-clim experiments, and acknowledges computational resources from the National Computational Infrastructure (NCI), and support from the Australian Government's National Environmental Science Program.

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

# Appendix A

## A1 Additional experiments under discussion for Tier 3

The experiments to be performed for Tier 3 have not been determined yet. The outcome from experiments in Tiers 1 and 2 are informing the decision process. Some experiments currently under discussion are briefly summarized below.

TBI-pace-X-clim

Where X stands for P, A, or I. Similar to TBI-pace-X, but restores to observed climatology in the basin of interest. This could serve as an additional reference to the TBI-pace-X experiments.

TBI-pace-X-clim-mod

Like TBI-pace-X-clim but restores to model climatology. These experiments have been performed with the ACCESS-CM2 model.

TBI-pace-AI

Restore the Atlantic and Indian Oceans simultaneously to study their combined effect.

976  TBI-pace-Pwedge

Similar to the TBI-pace-P but gradually narrows the restoring region toward the western Pacific, resulting in wedge that is centered on the equator, like the restoring region used by Kosaka and Xie (2013). This avoids restoring in the northwestern tropical Pacific, a region which may host variability distinct from ENSO.

981  TBI-pace-X20

Like TBI-pace-X but widens the restoring region to 20S-20N, with linear tapering to 30S and 30N. This would test the remote influence of subtropical SST anomalies.

985  TBI-hind-X20

Like TBI-hind-X, but widens the restoring region to 20S-20N, with linear tapering to 30S and 30N.

988  TBI-pace-X-1d

Like TBI-pace-X but uses very strong SST restoring with a time scale of 1 day over a 50 m deep layer. This would test whether the restoring time scale plays a crucial role in the strength of remote impacts.

992  TBI-hind-X-1d

Like TBI-hind-X but uses very strong SST restoring with a time scale of 1 day over a 50 m deep layer.

**A2 Restoring fields**

The target for the SST restoring will be the CMIP6 amip SST boundary conditions available at https://esgf-node.llnl.gov/search/input4mips/ (variable tosbcs). The current version is 1.1.9, which extends to December 2022. Please use this version. These monthly mean boundary conditions are centered on the middle of each month and should be linearly interpolated to the model time step. They are specifically modified such that the monthly mean observed value is recovered from the model output. See here for details: https://pcmdi.llnl.gov/report/pdf/60.pdf