# Peer review of "The Tropical Basin Interaction Model Intercomparison Project (TBIMIP)"

_EGUsphere, 2024_

## Author Comment (AC1)

**Response to Reviewer #1 (Michael Alexander)**

Thank you for your constructive and insightful comments. We hope that we have addressed them adequately. Below are our responses (in blue). If not otherwise mentioned, line numbers refer to the revised manuscript (without tracked changes).

**Major comments**

1) The manuscript could include a few figures from observations or previous experiments illustrating potential interactions and hypotheses to be explored, in addition to the schematic shown in Fig. 1.
We have added two figures (number 2 and 3 in the revised manuscript) that highlight the interdecadal modulation of the equatorial Atlantic influence on ENSO, and the influence of the northern tropical Atlantic on ENSO. In addition, Fig. 4 (previous Fig. 2) illustrates the inconsistent influence of ENSO on the equatorial Atlantic.

2) For many in the oceanography community "hindcast" is used to describe long simulations driven by atmospheric reanalysis (and ocean reanalyses) for regional models. (This is called a historical simulation here.) You might choose to use "re-forecasts" instead of "hindcasts" or add a sentence or two explaining how "hindcast" is being used in this context.
We have added a note regarding our use of the term "hindcast": "We note that we use "hindcast" in the sense of "reforecast", i.e. seasonal prediction experiments that are initialized from past observations."

3) Will the tapering method as a function of latitude (e.g., linear decrease with latitude) be prescribed to be the same across all experiments?
Yes, linear tapering will be used across all experiments. This was mentioned in line 201 of the original manuscript (now l. 234).

4) Can an explanation be provided for why the start of the tapering latitude is different in the Atlantic compared with the other two basins.
The choice was made based on the narrower region of deep convection in the tropical Atlantic, and the fact that previous studies have found unrealistic fluxes when SST restoring was used in the northern subtropical Atlantic (Kim et al. 2020; O'Reilly et al. 2023; Kim et al. 2024). This was already discussed in section 4.3 (now ll. 363-366). In addition, we now reference this explanation when we first mention the restoring regions in section 3 (ll. 233-234).

5) lines 217-218: States: "The technique for initializing the hindcasts (data assimilation etc.) is left to the modelling groups."  This could lead to major differences between the hindcasts (re-forecasts) especially in the first couple of months. Perhaps some tests with a single modeling system could be performed to investigate how much different initialization methods influence the forecast spread and perhaps how long it took for initialization differences not to have a notable influence on the re-forecasts (in a probabilistic sense).
The original idea was to require SST nudging as the initialization method for the reforecasts, but this would have meant additional effort and simulations, as most groups use some kind of 3D

data assimilation. It was therefore decided to let each group choose their own initialization method.

The SINTEX-F2 model uses both 3DVAR data assimilation and SST nudging for forecast initialization (12 ensemble members each). We have calculated the anomaly correlation coefficient (ACC) and the spread (standard deviation of the ensemble members) for the reforecast period 1991-2020 (Figs. R1 and R2, respectively). There is a systematic ACC increase in the eastern tropical Pacific and northern tropical Atlantic for the 3DVAR ensemble (Fig. R1), and systematic spread decrease in the equatorial Pacific (Fig. R2).

The differences described above increase with lead time, indicating that the initialization method has a lasting impact in some regions. Nevertheless, what is important for our purposes, is the changes relative to the control reforecast. We believe that these relative changes should not be too sensitive to the initialization method. We cannot rule out, however, that there is a systematic impact, and will try to investigate this in future experiments, potentially as part of the Tier 3 experiments. This is now mentioned in the revised manuscript (ll. 254-256).

[Figure]

**Figure R1.** ACC of SST for predictions initalized on February 1 for FMA (left column), MJJ (center column), and ASO (right column). The top row shows the ensemble mean of the predictions initialized with SST-nudging, the middle row shows the ensemble mean of the 3DVAR-intialized predictions, and the bottom row the difference between the two.

[Figure]

**Figure R2.** Like R1, but for inter-ensemble standard deviation (spread).

6) Lines 250-260 state:
"The top ocean level interacts with the atmospheric model component through a coupler routine (e.g., Craig et al. 2017), which regulates the exchange of fluxes between the atmosphere and ocean. Another approach for modifying SSTs is therefore through manipulating inside the coupler routine the heat flux that goes into the ocean, which is the method recommended for the TBIMIP experiments. The heat flux in tropical regions consists of four components: net surface

shortwave radiation, net surface longwave radiation, latent heat flux, and sensible heat flux. Of these, the sensible heat flux is usually chosen for manipulation (e.g., Kosaka and Xie 2013), and this is the method recommended for TBIMIP. Finally, because the flux coupler controls the SSTs that are "seen" by the atmospheric component, one can modify only this value, thereby "tricking" the atmosphere into reacting to a temperature that is different from the actual ocean SST. This approach leaves the ocean component completely unchanged (Richter and Doi 2019). Furthermore, it allows the SSTs to exactly follow a given distribution (as far as the atmosphere is concerned), rather than approximating it through correction terms. A potential drawback is that this can lead to very unrealistic heat fluxes into the atmosphere (Wang et al. 2005)."
And then on lines 281-282:
"Because the heat flux is absorbed in the top layer first, the immediate temperature response could lead to unrealistic changes in vertical stability"
These two statements seem contradictory, the top implying that you are not actually changing the ocean but just tricking it to see the altered state and the latter indicating an actual change in the ocean. Please clarify.

We have edited the manuscript to clarify that section 4.2 describes three different methods for constraining SSTs (ll. 282-283), and that the recommended method is through an additional heat flux term (method 2). As stated at the top of section 4.3, the discussion in this section only concerns method 2.

7) Lines 359-361: State "The curves essentially collapse into one, suggesting that the bias of a given model is mostly time-invariant. We conclude that using a shorter base period should not lead to major imbalances though this should be carefully evaluated for each model."
It may be worth exploring the results described in the paper:
Beverley, J.D., Newman, M. & Hoell, A. Climate model trend errors are evident in seasonal forecasts at short leads. *npj Clim Atmos Sci* 7, **285** (2024). https://doi.org/10.1038/s41612-024-00832-w

We have modified this statement to reflect the trend error at the beginning of the 21st century (ll. 404-406). The study by Beverley et al. (2024) is now cited, along with Kosaka and Xie (2013) and Wills et al. (2022).

8) Additional Tier 3 Experiments. The paper discusses a number of potential Tier 3 (optional) experiments using a hierarchy of models. Several of the proposed experiments are interesting and could be run relatively inexpensively. Here are some additional ones the project could consider:

- Use LIM or other methods to remove ENSO's (or other modes) impact on the observed SST anomalies in the other basins. The SST anomalies that are damped towards would remove this impact on the SST anomalies in the other basins and use those adjusted anomalies in either the historical or hindcast simulations. For example, the impact of ENSO on the tropical Atlantic could be estimated from observations and that part of the anomaly signal removed from the observed SST anomalies that are used in the TBI-PACE-AANOM experiment.
- Specify the observed winds or wind anomalies added to the model's climatological winds in the forcing regions rather than the SST (or SSTA). Since the oceans are primarily driven by winds in the tropics, both by the surface heat fluxes and dynamics (Ekman, upwelling, etc.). This might reduce or nearly eliminate the heat imbalance by

relaxing the heat into the ocean (although other issues might arise). A similar experiment design was used in

Ding, H., R. J. Greatbatch, M. Latif, W. Park, and R. Gerdes, 2013: Hindcast of the 1976/77 and 1998/99 Climate Shifts in the Pacific. J.                Climate, 26, 7650–7661, https://doi.org/10.1175/JCLI-D-12-00626.1.

- Base the temperature restoring term on the anomalous heat flux convergence in the ocean obtained from ocean reanalyses to estimate the ocean driven SST variability that is communicated to other basins.

Thank you for the valuable suggestions. We will consider those in our future discussions.

**Minor comments**

1) line 46: I suggest not using the colloquial expression "players" on line 46.  Perhaps "processes" instead.
Changed to "processes".

2) Lines 150-151: Suggest changing "a wealth of intercomparisons has been performed"  to "a wide-range of intercomparisons have been performed"
Done.

---

## Author Comment (AC2)

**Reviewer #2**

We thank the reviewer for their constructive and insightful comments and hope that we have addressed them adequately. Below are our responses (in blue).

The proposed pacemaker runs are designed to explore TBI via the atmospheric Walker circulation with SST restoration applied to narrow equatorial oceans (Fig. 3). Here are some slight changes in the domain(s) of SST restoration that could allow other modes of TBI.

1. Western North Pacific. Recent studies identified a coupled Indo-western Pacific ocean-atmosphere mode in post-ENSO summer (JJA). Its SST signature is also known as, but more complex than, the IOB mode, while the atmospheric component features a large-scale anomalous anticyclone (AAC) that covers the entire Indo-western Pacific north of the equator. A tropical Pacific domain with a wedge that reaches all the way to the maritime continent on the equator but avoids much of the off-equatorial western Pacific would be able to capture AAC and Asian summer monsoon variability/predictability (P. Zhang et al. 2024, J Climate). Climatically, the South China Sea is in the monsoon westerly regime that includes the North Indian Ocean (e.g., Fig. 2 of Zhang et al. 2024. Thus, the South China Sea should be part of the Indian Ocean, rather than the Pacific, for SST restoring.

2. In the Atlantic, the proposed SST restoration is limited to a narrow 10S-10N band. This could miss the subtropical atmospheric Rossby wave pathway connecting the broad tropical North Atlantic with the subtropical Northeast Pacific and then ENSO through WES (Ham et al. 2013a,b, cited in the paper) and low cloud (A. Miyamoto 2025, J Clim) feedback mechanisms.

The TBI team might want to consider these slightly modified configurations as possible extensions of the current TBIMIP.

Thank you for the thoughtful comments. These are valid concerns and, should computing resources allow it, we will try to address them with additional experiments. For the TBIMIP, we use the basin masks provided by the World Ocean Atlas (Locarnini et al. 2010), but an experiment using a wedge mask that excludes the South Chiane Sea and western Pacific is under discussion (see Appendix A TBI-pace-Pwedge).

Regarding the northern tropical Atlantic influence, Fig. 1 in Ham et al. (2013a) suggests that the convective response to the northern tropical Atlantic SST anomalies mostly occurs in the equatorial region. There are, however some precipitation anomalies north of 10N in the Caribbean during summer, so it cannot be ruled out that SST anomalies north of 10N can also have remote impacts. This should be further investigated.

**Minor comments**

There are two Chang et al. (2006) references, which may need to be labeled as a, b.
Done.

Xie and Carton (2004) not in References.

We added this reference.

L289. "subtropical" --> "subpolar"?
"Subtropical" is correct but we realized that one of the references (Kim et al. 2024) is not relevant here and have removed it. In addition, another reference (Kim et al. 2020) only indirectly concludes that the role of the subtropical North Atlantic may have been overestimated. We have reworded this sentence accordingly.

---

## Author Comment (AC3)

**Response to Editor (Penelope Maher)**

We thank you and the reviewers for all the helpful comments and hope that we have addressed them adequately. Below are our responses to the editorial comments (in blue).

In papers which describe the methodology of a new MIP, all the relevant technical details should be in the manuscript or supplementary. I would recommend you include all relevant technical information in the manuscript, and any additional information which is helpful be added to the supplementary (as it currently stands there is repetition between the full description and the manuscript). This will help people who may wish to contribute to the MIP. Further, this means the full method has undergone peer review, rather than the archive which has not technically been peer reviewed as part of the manuscript.
We have inserted all relevant information into the manuscript (section 3, section 5 and Appendix A).

Tier 3 experiments in Table 1. There was a bit of a disconnect here between the manuscript (which does not explain Tier 3 experiments other than to say they are planned) and the full description in the archive (which suggests possible experiments). Personally I think it would be clearer to only describe Tiers 1-2 in the table and then have a section description of possible directions for further work for the Tier 3 experiments (which would presumably be part of TBIMIP2 or are you planning as part of TRIMIP? ).
We believe it is important to mention in the main text that additional experiments are under consideration. These experiments are expected to be part of TBIMIP. There are currently no plans for a TBIMIP2.
We now mention that some suggestions for additional experiments are given in the Appendix.
We have also deleted the mentioning of Tier 3 experiments in Table 2 (now Table 3).

Optional editing to the manuscript

 - Fig 1: please include units (it is obvious but best practice to include).
Done.

 - L99: The proxy record are important parts of the observational record and I find the comment "This makes paleo proxies subject to uncertainties and inhomogeneities" a little dismissive. All datasets and models we use are subject to uncertainties so perhaps rewording to ",  which can contribute to uncertainties."?
Changed.
This passage discusses the strengths and weaknesses of observations, model simulations, and paleo proxies. There was no intent to be particularly dismissive of any particular approach.

 - Table 1: The ESGF database use lowercase for all experiment names. I suggest you follow this approach in the manuscript.

We have changed the experiment names to lowercase, except for the TBI acronym and the letters designating the three basins (P, A, I). Mixed lower/upper case experiment names are quite common in CMIP (e.g., "1pctCO2", "hist-resAMO").

 - Take care with statements like the following which may not get updated and are therefore not encouraged: "This is IPRC publication X and SOEST contribution Y."
It is an institutional requirement for co-author Malte Stuecker to add this statement to any publication. These numbers are issued upon acceptance of the manuscript and are updated during the proofing stage. We have never encountered any issues with this procedure and hope that GMD can accommodate these requirements.

---

## Author Response (AR2)

**Response to Editor (Penelope Maher)**

We thank you for the careful checking of our manuscript and your comments. We hope to have addressed them adequately with our responses (below in blue).

1. Adding new authors is a little uncommon at this stage. Could you comment on why they have been added now and were not at initial submission?
We added Ping-Gin Chiu and Takeshi Doi as co-authors because their contributions to performing relevant simulations used in designing the protocol had been overlooked in the initial submission. In addition, co-author Doi contributed during the revision stage by exploring the influence of 3DVAR vs. SST-nudging initializations.

2. During validation, this was picked up: The ROR database lists the institution of the corresponding author but with a different city than given in the manuscript. Please clarify whether the ROR "Japan Agency for Marine-Earth Science and Technology (Yokosuka, Japan)" is still correct.
JAMSTEC is headquartered in Yokosuka but maintains several offices in Japan, one of them being located in Yokohama, where I. Richter, T. Doi and T. Kataoka are based. Unfortunately, the Research Organization Registry (ROR) only lists the Yokosuka headquarters. We therefore selected this location from the pulldown menu.

3. The 2 new figures and new table (Fig 2-3, Table2) all have introduction material that would be better in the main text and not in captions.
We have moved part of the captions for Table 2 and Fig. 3 to the main text.
The purpose of Figs. 2 and 3 is to illustrate hypothesized interactions and outstanding issues, as suggested by reviewer #1. As such, the details of those figures are not essential to the main text and would disrupt the flow of the introduction. We have therefore decided to leave some of these details in the captions.

4. Fig 5: does the yellow shading around the red have any meaning? If it is just to show the buffer, then could it be a low opaque red please?
We have replotted the figure using a different color scheme with monochrome opacity gradients.

5. Please thank your reviewers in the acknowledgement section.
Done.

---

## Author Response (AR3)

Ingo Richter
APL, JAMSTEC
3173-25 Showa-machi, Kanazawa-ku
Yokohama, 236-0001
Tel.: +81-45-778-5523
E-mail: richter@jamstec.go.jp

25 February 2025
To: Dr. Penelope Maher
RE: Files for journal article production of manuscript egusphere-2024-3110

Dear Dr. Maher,
We thank you very much for your positive evaluation and the careful checking of our manuscript. We are herewith uploading our files for article production. The manuscript is the same as the accepted version, except that in the acknowledgments we have replaced the placeholders X and Y for the IPRC and SOEST contributions, respectively, with actual numbers.
Best regards,
Ingo Richter